# Evaluation of the Oral Microbiome before and after Treatments for Halitosis with Photodynamic Therapy and Probiotics—Pilot Study

**DOI:** 10.3390/healthcare12111123

**Published:** 2024-05-30

**Authors:** Pamella de Barros Motta, Marcela Leticia Leal Gonçalves, Juliana Maria Altavista Sagretti Gallo, Ana Paula Taboada Sobral, Lara Jansiski Motta, Elaine Marcílio Santos, David José Casimiro de Andrade, Cinthya Cosme Gutierrez Duran, Kristianne Porta Santos Fernandes, Raquel Agnelli Mesquita-Ferrari, Anna Carolina Ratto Tempestini Horliana, Sandra Kalil Bussadori

**Affiliations:** 1Post Graduation Program in Biophotonics Applied to Health Sciences, Universidade Nove de Julho (UNINOVE), São Paulo 01525-000, SP, Brazil; pamellabmotta@gmail.com (P.d.B.M.); larajmotta@terra.com.br (L.J.M.); cinthya.cgduran@gmail.com (C.C.G.D.); kristianneporta@gmail.com (K.P.S.F.); raquel.mesquita@gmail.com (R.A.M.-F.); annacrth@gmail.com (A.C.R.T.H.); 2Post Graduation Program in Health and Environment, Universidade Metropolitana de Santos (UNIMES), Santos 11045-002, SP, Brazil; marcelalleal@hotmail.com (M.L.L.G.); anapaula@taboada.com.br (A.P.T.S.); elaine.marcilio@unimes.br (E.M.S.); 3Post Graduation Program in Veterinary Medicine in the Coastal Environment, Universidade Metropolitana de Santos (UNIMES), Santos 11045-002, SP, Brazil; jualtavistagallo@gmail.com; 4School of Dentistry, University of Porto, 4099-002 Porto, Portugal; casimiroandrade@gmail.com; 5Dentistry College, Universidade Metropolitana de Santos (UNIMES), Santos 11045-002, SP, Brazil

**Keywords:** halitosis, microbiome, antimicrobial photodynamic therapy, probiotics

## Abstract

Background: To compare photodynamic therapy and the use of probiotics in reducing halitosis assessed through gas chromatography and microbiome analysis. Methods: Participants aged from 18 to 25 years showing sulfide (SH2) ≥ 112 ppb on gas chromatography were selected. They were divided into four treatment groups: Group 1—Tongue Scraping; Group 2—Antimicrobial Photodynamic Therapy (aPDT); Group 3—Probiotics; and Group 4—Antimicrobial Photodynamic Therapy (aPDT) and Probiotics. The halimetry process was performed before, immediately after the treatments, and 7 days, 14 days, and 30 days after the initial collection. The collections for later microbiological analysis were made along with the halimetry for microbiome analysis. Results: Treatment with aPDT or probiotics under these experimental conditions was not able to change the bacteria present in the biofilm of the tongue. Conclusions: More research is needed to know the behavior of the oral microbiome in the presence of halitosis and the effectiveness of new treatments.

## 1. Introduction

Halitosis is characterized by an offensive and unpleasant odor originating from the oral cavity and nasopharynx [1]. This malodor mainly results from the presence of unpleasant smell substances—known as volatile sulfur compounds (VSCs)—which are gases present in the breath air that are produced by oral bacteria on substrates containing sulfur. The main compounds associated with halitosis are sulfide (H2S) (related to tongue coating), dimethylsulfide ((CH3)2S) (related to periodontal pockets), and methylmercaptan (CH3SH) (related to systemic alterations). These compounds are produced by Gram-negative anaerobic bacteria [2]. Epidemiological studies report the prevalence of halitosis can vary between 2.4 and 78%, depending mostly on the assessment method (reports or objective measurement of gases) [3]. This condition has great social implications because it brings embarrassment to its bearer [4].

The oral microbiome is made up of a huge variety of microorganisms. The terms “microbiota” and “microbiome” are often used interchangeably, however, there are differences in their definitions. The microbiota refers to the living microorganisms themselves present in each environment, such as the oral and intestinal microbiota. The term microbiome refers to the grouping of genomes from the entire environment, that is, in addition to living microorganisms, it includes structural elements as well as environmental conditions and metabolites. [5]. When there is balance and harmony between the microbiome and the host, there is health, that is, the microorganisms contribute positively to the well-being of the host. Habits in general (food, stress, use of tobacco and other drugs, alcoholic beverages) directly interfere with the relationship between the host and the oral microbiome. This means that these habits can alter the composition of the microbiome in such a way that an imbalance in this ecosystem begins [6]. When there is disharmony, there are diseases such as halitosis, caries, and periodontal disease, among others [7]. The balance in the microbiome is essential for oral health, which in turn is essential for the general health of the host.

Ye et al. reported that people with halitosis have a more diverse microbiome than those without halitosis. According to the authors, the main bacteria related to halitosis are *Prevotella*, *Alloprevotella*, *Leptotrichia*, *Peptostreptococcus*, and *Stomatobaculum* [8]. Patients with halitosis apparently have a greater bacterial diversity than control patients. There are 13 phyla, 23 classes, 37 orders, 134 genera, 266 species, and 349 operational taxonomic units that make up the microbial communities present in this diversity [8,9]. With the advent of sequencing, these new genera of bacteria related to this condition are being studied. 

There are three main methods for the diagnosis of halitosis. The most common consists of a subjective method called the organoleptic test. In this test, the patient exhales air close to the evaluator who quantifies the bad breath through a score. Some factors must be considered to contraindicate the use of the organoleptic test, such as the risk of contamination by SARS-CoV-2, for example. [10]. Another method that can be used is the use of portable breath meters. The use of this equipment was evaluated and compared to the organoleptic test and showed high sensitivity and specificity, being a useful and practical instrument for the detection of halitosis. Finally, gas chromatography is the most suitable test for diagnosing the presence and type of halitosis by qualitatively and quantitatively analyzing volatile sulfur compounds [11,12,13]. 

Halitosis treatments are based on controlling and disorganizing biofilms rich in bacteria related to the production of VSC. Among them is the use of antimicrobial substances, such as chlorhexidine (CHX), cetylpyridinium chloride (CPC), and triclosan, contained in products for oral hygiene such as toothpaste [3], antimicrobial photodynamic therapy [11,14,15,16,17], and probiotic therapies [18]. Antimicrobial photodynamic therapy (aPDT) has been commonly used in oral health treatments, including for the treatment of halitosis. This approach involves the use of a visible light source (laser or LED) and a compatible photosensitizer. Reactive oxygen species are formed by the interaction of light with the photosensitizer in the presence of oxygen, causing the cell death of microorganisms [16]. 

Because it is a non-invasive technique that does not cause aftereffects, aPDT has been used as an alternative or adjuvant to conventional antimicrobial treatments. Treatments with probiotics consist of the administration of non-pathogenic live microorganisms that aim to enhance the equilibrium of the microbiome [19]. Therefore, the objective of this study was to compare photodynam1ic therapy and the use of probiotics in reducing halitosis assessed through gas chromatography and microbiome analysis. The challenge of this study was also its main limitation: the number of patients (as it is a pilot study) and the lack of control over the correct oral hygiene of effective patients at home, despite them having received an explanation of the terms and a folder with information and guidance on hygiene oral.

## 2. Materials and Methods

Six participants were selected, according to inclusion criteria: participants of both genders, aged from 18 to 25 years, and showing halitosis, defined as gas chromatography sulfide (SH2) ≥ 112 ppb in the OralChromaTM device. Exclusion criteria were individuals with dentofacial anomalies (such as cleft lip, palatine, or nasopalatine fissures), in orthodontic or orthopedic treatment, in oncological treatment, with any health problems (gastrointestinal, renal, hepatic), being treated with antibiotics up to 1 month before the survey and pregnant women. The study was approved by the Ethics Committee of Universidade Nove de Julho (UNINOVE), under process number 3,669,442 and all participants signed an informed consent form. 

Participants were instructed, through a lecture and digital files, to brush with amine fluoride toothpaste (Elmex^®^) and to floss 3 times a day after meals for 30 days. They were taught how to perform the Bass technique, in which the bristles of the brush are positioned at an angle of approximately 45° towards the inside of the gingival sulcus, both on the free and proximal surfaces, in addition to short and slightly circular vibrating movements. After the oral hygiene instruction, the initial assessment of the tongue coating proposed by Shimizu et al. was carried out using the Coated Tongue Index (CTI) [20]. For this evaluation, the tongue is divided into 9 sectors, and each sector receives a score, being 0—no coating, 1—coating allowing the visualization of the papillae, and 2—thick coating not allowing the visualization of the papillae. These grades were added, divided by 18, and multiplied by 100 to obtain a final index of 100%. It should be clarified that the participants were only instructed and did not brush and floss in the same treatment session. Afterward, the evaluation was made by gas chromatography with the OralChromaTM device, and the microbiological collection was performed for later evaluation of the microbiome. 

The collection of oral air followed the manufacturer’s instructions (Oral ChromaTM Manual Instruction), in which the subject was required to wash out his mouth with cysteine (10 mM) for 1 min, then keep their mouth closed for another 1 min. A syringe from the same manufacturer designed to collect oral air was then introduced into the subject’s mouth. The subjects kept their mouth closed for a further minute, respiring through the nose, without touching the syringe with the tongue. The plunger was then pulled out, pushed in, and pulled out again to fill the syringe with the breath sample. The gas injection needle was placed on the syringe, and the plunger was adjusted to 0.5 mL. The collected gases were injected into the inlet port of the device with a single movement. To avoid changes in halimetry, subjects were instructed to follow the following guidelines: Forty-eight hours before the assessment, do not eat spicy foods (garlic, onion), do not drink alcohol, and do not use breath freshener. On the day, eat up to 2 h before the assessment. Do not consume coffee, candies, or chewing gum, and do not use oral and personal hygiene products (deodorant, perfumes, creams); in addition, brushing should be performed with water only. The halimetry process with OralChromaTM was performed before, immediately after the treatments, and 7 days, 14 days, and 30 days after the initial collection, depending on the different treatments. 

The collections for later microbiological analysis were made along with the halimetry. 

A sterile swab was used to collect the tongue coating by passing it on the dorsum of the tongue, performing one backward and forward movement. The samples were deposited in sterile tubes containing Tris–EDTA buffer (10 mM Tris–HCL, 0.1 mM EDTA, pH 7.5), identified, and stored at −80 °C until analyzed. Samples were frozen due to the impossibility of performing all microbiome analyses in a single day. 

In the microbiome procedure, all sequencing, raw data collection, and analysis were executed by the ByMyCell laboratory. DNA extraction was performed using the DNeasy PowerSoil Pro Kit (Qiagen^®^, Hilden, Germany). The resulting fragments were submitted to sequencing of the V3–V4 region of the 16SrRNA gene on the Nanopore platform. After processing reads and removing chimeras, an average of 10,500 reads per sample remained. After filtering, the reads were classified taxonomically, using the SILVA 123 database, obtaining the classification of 414 genera and 901 species. 

Participants received different proposed treatments for halitosis from tongue coating, according to the descriptions below. 

Group 1—Tongue Scraping

In one individual participant (1), tongue scraping was performed with a plastic scraper. The lingual coating was removed using the scraper on the back of the tongue with ten posteroanterior movements until the scraper came out clean of the surface.

Group 2—Antimicrobial Photodynamic Therapy (aPDT)

In another individual participant (28), one session of aPDT was carried out with an LED photopolymerizing device—Valo Cordless Ultradent^®^ (South Jordan, Utah, USA)—with coupled radiometer, spectrum of 440–480 nm, and irradiance of 450 mW/cm and with 5 sprays of photosensitizer (PS) annatto (manipulated at a concentration of 20% (Formula e Ação^®^, São Paulo, Brazil)), covering the middle third and dorsum of the tongue in spray, the pre-irradiation time was 2 min. The surplus was removed using a sucker to keep the surface moist with the PS itself, without using water. Six points were irradiated with 1 cm between points, considering the halo of light scattering and the effectiveness of aPDT. The apparatus was previously calibrated with a wavelength of 395–480 nm, for 20 s per point, the energy level was set to 9.6 J, and the light was irradiated so that a halo of 2 cm in diameter per point was formed [18,21]. 

Group 3—Probiotics

Two participants (35 and 39) were instructed to ingest probiotic capsules. Pharmacy compounded capsules containing strains of Lactobacillus salivarius WB21 (6.7 × 108 CFU) and xylitol (280 mg) were used. Forty-two capsules were given to each patient, who had to take 1 capsule, 3 times a day after meals, for 14 days. 

Group 4—Antimicrobial Photodynamic Therapy (aPDT) and Probiotics

Two participants (6 and 18) received both the aPDT and probiotic treatments, as described above.

To study abundance, the Kruskal–Wallis Test with Dunn’s post hoc test was used (Software Rstudio 2022.07.0 Build 548© 2009–2022 RStudio, PBC. The packages used were dbplyr, rstatix, ggplot2, https://posit.co/download/rstudio-desktop/, accessed on 19 June 2020).

For the analysis of the microbiome of the tongue coating, the analysis of alpha diversity was performed. 

The microbiota raw sequencing data were submitted to the NCBI (SUB14281977).

## 3. Results

Table 1 shows the results of the initial tongue coating index and the amount of sulfide in ppb in each halimetry test. In those treated with scraper and aPDT, the measurements were taken at the initial times, immediately after, 7 days after, and 30 days after, for control. In those treated with probiotics, the initial times, and 7 days, 14 days, and 30 days after were performed. In these participants, it was not possible to carry out the “immediately after” time, since the participant had to start ingesting the probiotics. Consequently, the 14-day control period was added, as the participant ingested the capsules for 14 days. In participants treated with both aPDT and probiotics, all times were performed (initial, immediately after, and 7 days, 14 days, and 30 days after). 

### Microbiome

For the statistical analysis, the groups in Table 2 were considered.

For the analysis of the microbiome of the tongue coating, the analysis of alpha diversity was performed. It can be observed that there was no difference between the groups by the analysis of Chao1, Shannon, and Simpson. 

In Figure 1, Figure 2 and Figure 3, the comparison between times of the analyzed groups to verify the alpha diversity are shown. 

As for the relative abundance analysis regarding the genera found, a difference was observed only for the genus Pseudarthrobacter (*p* < 0.05) between Group 2 and Group 3 at 14 days. In Figure 4, we can see the 20 most abundant genera present in the analyzed samples. 

In the Venn diagram represented in Figure 5, we can observe that 12 genera were common among the Halitosis, Scraping, and aPDT groups.

In the Venn diagram represented in Figure 6, we can see that there was a decrease in the number of genera found when comparing the times immediately after treatment and 7 days. 

In the Venn diagram represented in Figure 7, we can see that there was similarity of 7 genera found at 7 days. 

In the Venn diagram represented in Figure 8, we can see that there was similarity of 14 genera found at 14 days. 

Regarding the relative abundance between the species present in the samples, in 25 generals, we can say that there was no difference between the groups (*p* > 0.05, Kruskal–Wallis test). Figure 9 shows the relative abundance of the 20 most abundant species present in the analyzed samples. 

In the Venn diagram represented in Figure 10, we can see species that were more abundant than 1% in control groups. 

In the Venn diagram represented in Figure 11, we can see that there was a decrease in the number of species found when comparing the times immediately after treatment with aPDT and 7 days. 

In the Venn diagram represented in Figure 12, we can see that there was similarity of 17 species found at 7 days. 

In the Venn diagram represented in Figure 13, we can see that there was similarity of 19 species found at 14 days. 

As for the prediction of metabolism, we can observe that there was no difference between the analyzed groups (*p* < 0.05) (Figure 14). 

## 4. Discussion

The oral microbiota consists of 50 to 100 billion bacteria [19,20,22,23,24,25,26]. In 1963, Socransky et al. suggested that only 50% of the oral cavity microbiota had been cultured [25].

Recent work emphasizes that the oral microbiota can reach the intestine and throughout the body through blood circulation, potentially leading to numerous systemic diseases. This infiltration occurs through the junctional epithelium below the gingival sulcus, which is connected to the cementum by the hemidesmosome, which is weaker than the desmosome, and, therefore, more permeable [25].

Dysbiosis of the oral microbiota is the primary etiological factor of halitosis. Instead of studying the pathogenicity of individual bacteria, research shifted to the study of the relationship between the composition of the oral microbiome and systemic diseases. Studies show that it is necessary to understand the specific mechanisms that regulate the balance of the oral microbiota for the development of prevention and treatment strategies for oral diseases and even systemic diseases [26].

It is known that molecular techniques are more suitable for testing and evaluating the microbiome of the oral cavity [19,20] with qPCR and 16S rRNA amplicon sequencing being the most used [27,28,29,30]. 

In order to obtain an overview of the tongue microbiota at an ecological level, more recent work has carried out the sequencing of the 16S rRNA amplicon. These studies have shown that there is a prevalence of many other species in the development of halitosis [2,27,28,31]. It seems clear that a bacterial community is responsible for maintaining halitosis, and its treatment remains a challenge. In a published review [28], the authors showed that the most prevalent genera in intraoral halitosis were *Aggregatibacter*, *Capnocytophaga*, *Campylobacter*, *Clostridiales*, *Dialister*, *Leptotrichia*, *Prevotella*, *Peptostreptococcus*, *Peptococcus*, *Parvimonas*, *Selenomonas*, *Treponema*, and *Tannerella.* Other authors found *Streptococcus*, *Veillonella*, *Gemella*, *Granulicatella*, *Neisseria*, *Haemophilus*, *Selenomonas*, *Fusobacterium*, *Leptotrichia*, *Prevotella*, *Porphyromonas*, and *Lachnoanaerobaculum* [2]. Another study [8] demonstrated that the genera *Prevotella*, *Alloprevotella*, *Leptotrichia*, *Peptostreptococcus*, and *Stomatobaculum* exhibited higher relative percentages in halitosis samples compared to healthy samples. In our work, the most abundant genera were *Streptococcus*, *Neisseria*, *Veillonella*, *Granulicatella*, *Oribacterium*, *Gemella*, *Rothia*, *Fusobacterium*, *Schaalia*, *Prevotella*, *Campylobacter*, *Lanchnoanaerobaculum*, *Porphyromonas*, *Solobacterium*, *Leptotrichia*, *Stomatobaculum*, *Haemophilus*, *Enterococcus*, and *Prevotellamassilia*. Therefore, these results corroborate those of other authors regarding genera: *Campylobacter* [28], *Leptotrichia* [8,28], *Prevotella* [2,8,28], *Streptococcus*, *Neisseria*, *Veillonella*, *Granulicatella Fusobacterium*, *Lanchnoanaerobaculum*, *Porphyromonas Haemophilus*, and *Gemella* [2,28]. 

As for the species, the most prevalent were *Neisseria perflava*, *Streptococcus salivarius*, *Streptococcus oralis*, *Veillonella parvula*, *Granulicatella adiacens*, *Streptococcus mitis, Granulicatella elegans*, *Rothia micilaginosa*, *Gemella sanguis*, *Streptococcus australis*, *Schaalia odontolytica, Oribacterium*, *asaccharolyticum*, *Streptococcus koreensis tobeensis*, *Veilon*, *vestibular Streptococcus*, *Streptococcus sublava*, *Fusobacterium periodonticum*, *campylobacter concisus*, and *Neisseria cirerea.* These results corroborate those of other studies regarding the species *Streptococcus mitis* [32,33], *Fusobacterium periodonticum*, [34] *Streptococcus oralis* [31], *Streptococcus salivarius* [31], and *Granulicatella elegans* [35], as these studies also link halitosis with the presence of these bacteria. The identification of possible bacteria associated with halitosis is essential to develop new strategies in the treatment of halitosis.

In 2021, Zhang et al. [1] conducted a study on the dynamism of the microbiota related to halitosis in children. The method of 16S rRNA gene sequencing was also used to reveal the shift in the tongue-coating microbiome in these children during a 12-month period. Halitosis-enriched species *Prevotella melaninogenica*, *Actinomyces sp._HMT_180*, and *Saccharibacteria TM7_G-1_bacterium_HMT_352* were finally selected as biomarkers in the halitosis-onset prediction model after screening, showing different types of species than the ones that were previously more researched. In this study, the microbiome composition and relative abundance of the tongue coatings in the halitosis and control groups differed remarkably, even prior to the onset of the clinical manifestations of halitosis during the 12 months of the trial. These results suggest that as a preventive measure, the tongue coating plate control instructions can be carried out prior to the onset of halitosis. This is an interesting result that the authors were able to obtain as they used a group without halitosis, which was not done in our study since we were testing treatment options.

Regarding the treatment options offered in our study, alternative options to conventional treatments were aPDT and probiotics. Several previous studies [11,14,15,16,17] demonstrated that in gas chromatography analysis, aPDT was able to reduce VSC levels immediately, although this clinical success was not demonstrated in the analysis of the microbiome performed in the present study. The aPDT technique that was used was based on previous protocols and clinical trial studies [18,21]. In clinical studies with results, these were in line with our results regarding halimetry, having only an immediate result. However, in these studies, unlike the present study, microbiological analyses were not performed. In 2019 [36], a systematic review was performed to summarize the evidence on the effect of probiotics on halitosis. Meta-analysis revealed that organoleptic assessment scores were significantly lower in subjects receiving probiotics than in placebo groups, but no significant difference was observed in VSC concentration, results like our sulfide and microbiome analysis. Another systematic review [21], carried out in 2022, pointed out that the Lactobacillus species, also used in this study, is the most proposed for the treatment of halitosis. Both reviews agree on the fact that the available evidence is insufficient for recommending probiotics for oral malodor, requiring further clinical studies, such as the present study, in this area. 

## 5. Conclusions

It can be concluded that treatment with aPDT or probiotics under these experimental conditions was not able to change the lingual coating microbiota of patients with halitosis. More research is needed to better understand the behavior of the oral microbiome in the presence of halitosis and the effectiveness of new treatments to be proposed.

## Figures and Tables

**Figure 1 healthcare-12-01123-f001:**
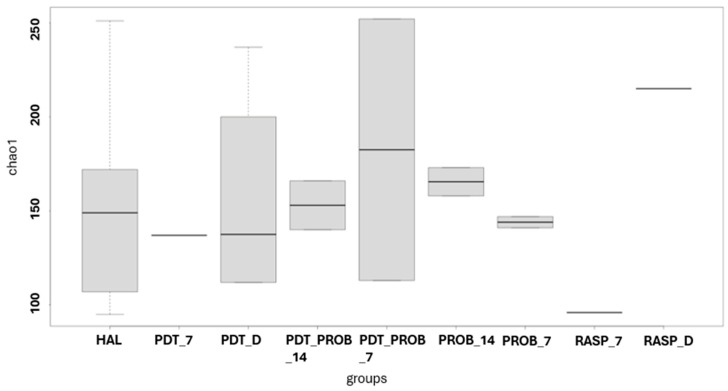
Comparison between times of the analyzed groups to verify the alpha diversity. RASP—Scraper Group, PDT—aPDT Group, PROB—Probiotics Group, PDT + PROB—aPDT + Probiotics Group. Times analyzed: D-immediately after treatment, 7—7 days, 14—14 days.

**Figure 2 healthcare-12-01123-f002:**
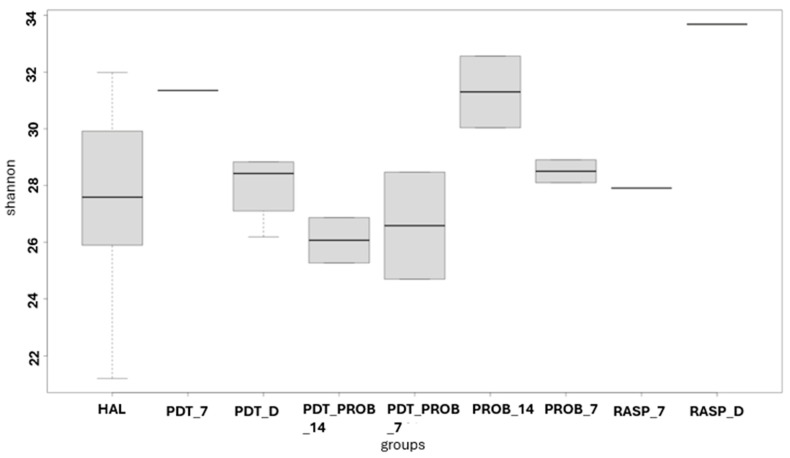
Comparison between times of the groups analyzed for verification of alpha diversity. RASP—Scraper Group, PDT—aPDT Group, PROB—Probiotics Group, PDT + PROB—aPDT + Probiotics Group. Times analyzed: D-immediately after treatment, 7—7 days, 14—14 days.

**Figure 3 healthcare-12-01123-f003:**
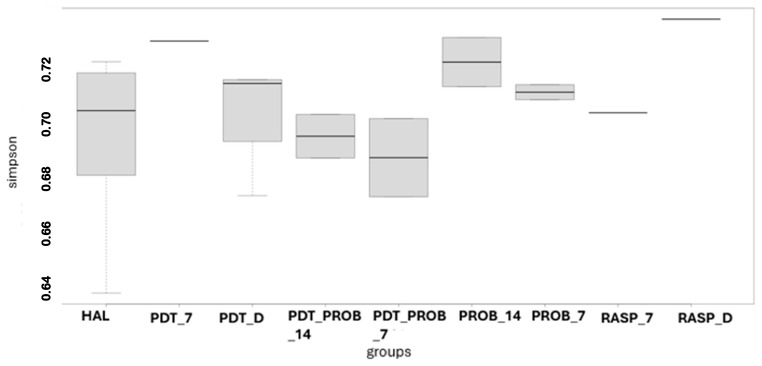
Comparison between times of the analyzed groups to verify the alpha diversity. RASP—Scraper Group, PDT—aPDT Group, PROB—Probiotics Group, PDT + PROB—aPDT + Probiotics Group. Times analyzed: D—immediately after treatment, 7—7 days, 14—14 days.

**Figure 4 healthcare-12-01123-f004:**
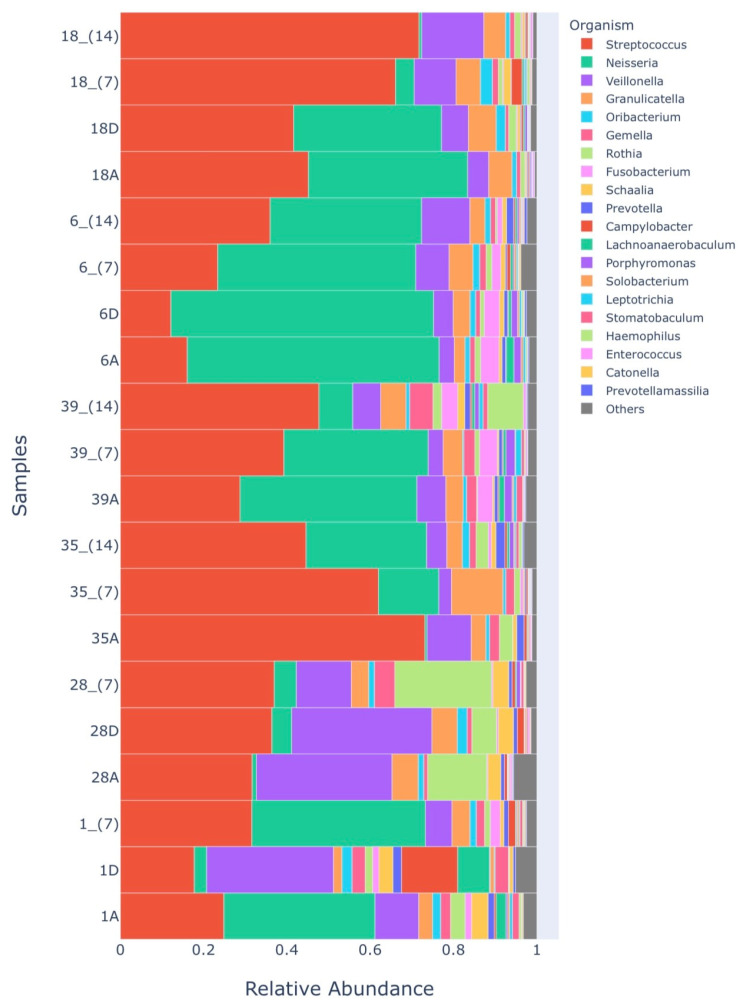
Diagram showing the relative abundance of the 20 most abundant genera present in the analyzed samples.

**Figure 5 healthcare-12-01123-f005:**
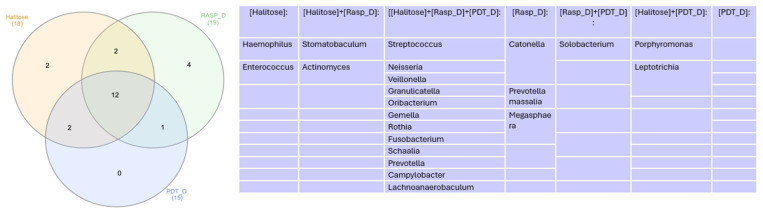
Venn diagram for genera more abundant than 1% in control groups.

**Figure 6 healthcare-12-01123-f006:**
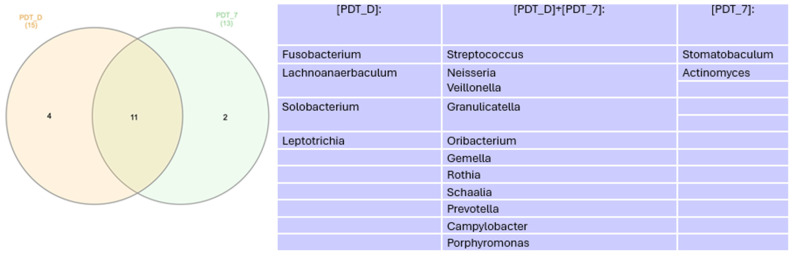
Venn diagram for genera more abundant than 1% at 7 days. PDT—aPDT group, PDT + PROB—aPDT group + probiotics.

**Figure 7 healthcare-12-01123-f007:**
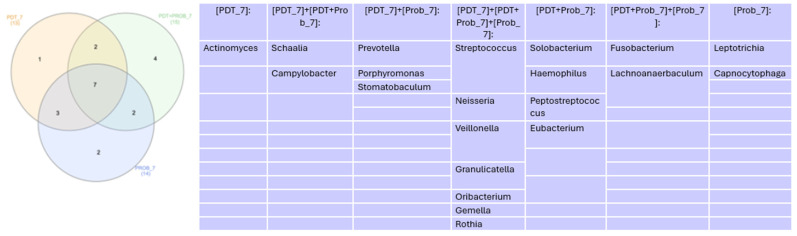
Venn diagram for genera more abundant than 1% at 7 days.

**Figure 8 healthcare-12-01123-f008:**
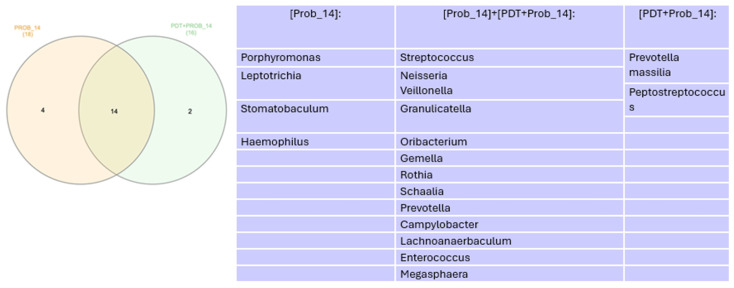
Venn diagram for genera more abundant than 1% at 14 days. Relative abundance (species).

**Figure 9 healthcare-12-01123-f009:**
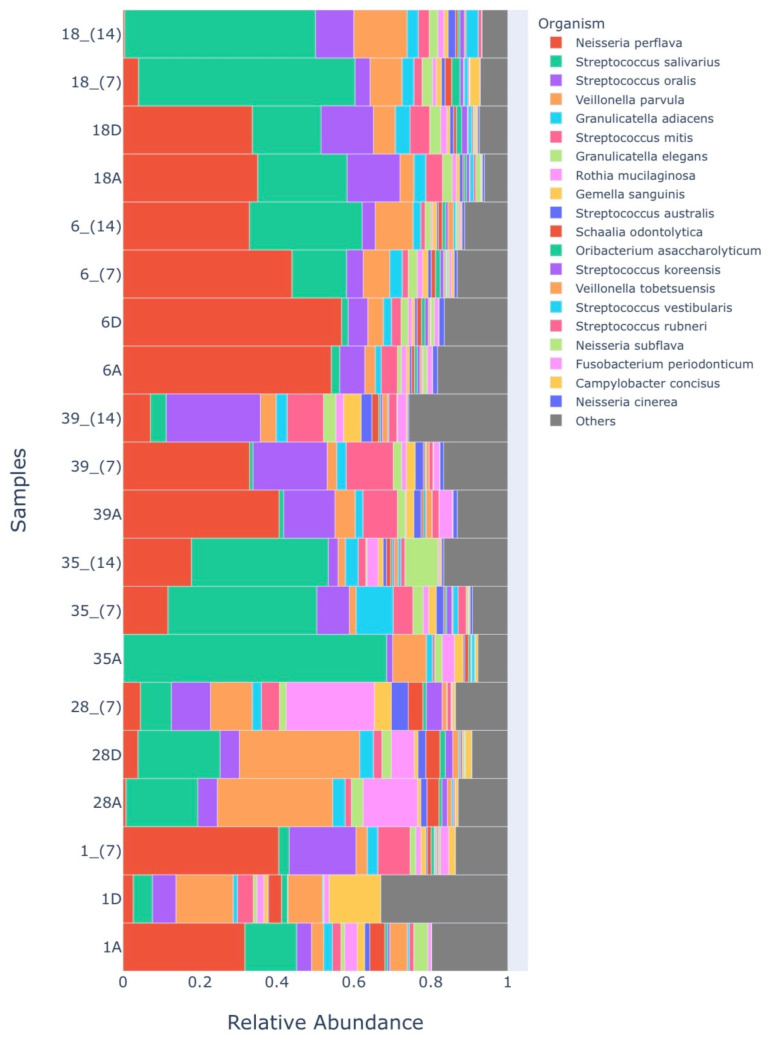
Diagram showing the relative abundance of the 20 most abundant species present in the analyzed samples. Group. Times analyzed: A—before starting treatment, D—immediately after treatment, 7—7 days, 14—14 days.

**Figure 10 healthcare-12-01123-f010:**
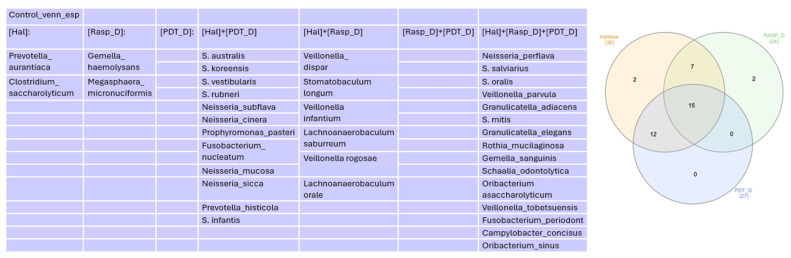
Venn diagram for species more abundant than 1% in control groups.

**Figure 11 healthcare-12-01123-f011:**
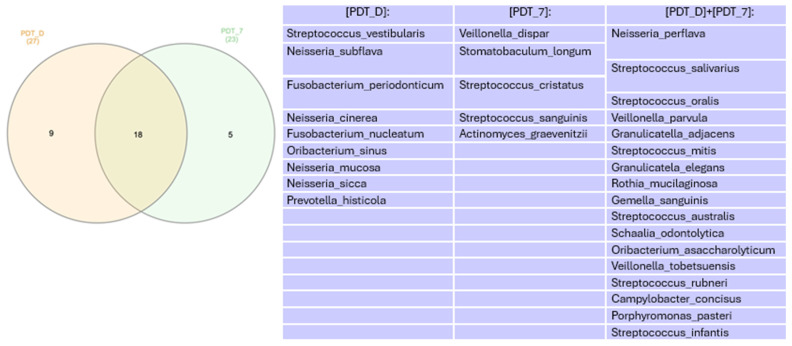
Venn diagram for species more abundant than 1% in the groups in which aPDT was performed.

**Figure 12 healthcare-12-01123-f012:**
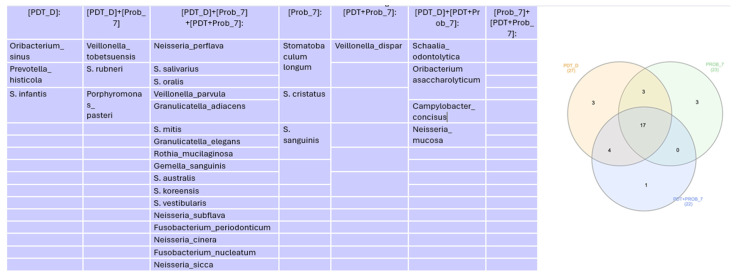
Venn diagram for species more abundant than 1% at 7 days. PDT—aPDT group, PDT + PROB—aPDT group + probiotics. Times analyzed: D—immediately after treatment, 7—7 days.

**Figure 13 healthcare-12-01123-f013:**
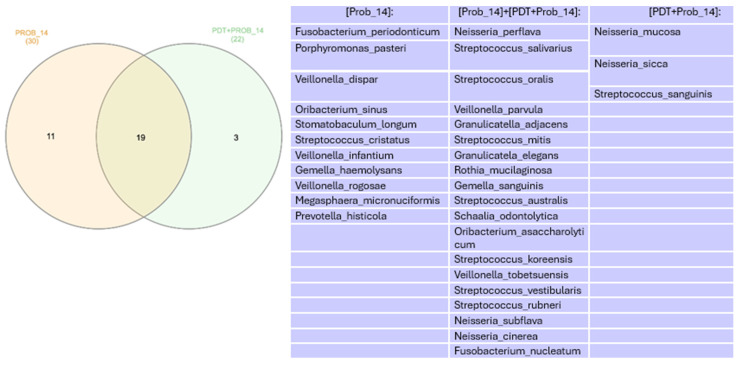
Venn diagram for species more abundant than 1% at 14 days. PDT—aPDT group, PDT + PROB—aPDT group + probiotics. 14—14 days.

**Figure 14 healthcare-12-01123-f014:**
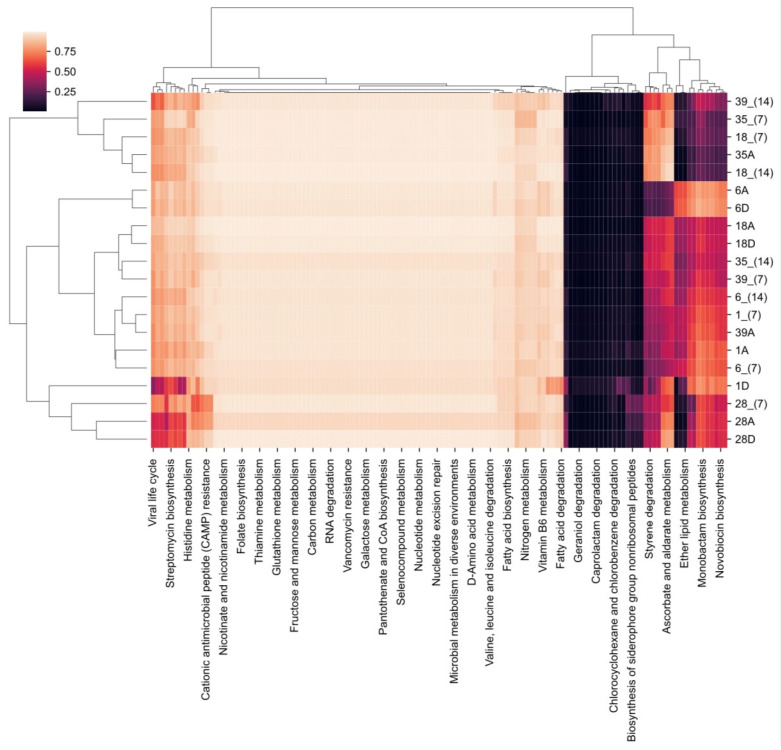
Schematic drawing showing the prediction of metabolism between the groups and times analyzed.

**Table 1 healthcare-12-01123-t001:** Performed treatments, initial Coating Tongue Index, and sulfide level the participants presented in all evaluation times.

Participant and Treatment That Was Performed	Coated Tong Index (CTI)	Sulfide in ppb—Initial	Sulfide in ppb— Immediately after Treatments	Sulfide in ppb—after 7 Days	Sulfide in ppb—after 14 Days	Sulfide in ppb—after 30 Days
1—Tongue scraping	16.66%	1436	0	592	-	1224
28—aPDT	50%	2175	7	1751	-	599
35—Probiotics	66.66%	1354	-	279	780	1648
39—Probiotics	66.66%	437	-	65	95	-
6 aPDT + Probiotics	16.66%	621	32	173	523	342
18 aPDT + Probiotics	16.66%	482	0	7	497	282

**Table 2 healthcare-12-01123-t002:** Groups and samples identification.

Groups Microbiome ^1^	Sample Identification
HALITOSE	1A
RASP_D	1D
RASP_7	1_ (7)
HALITOSE	28A
PDT_D	28D
PDT_7	28_ (7)
HALITOSE	35A
PROB_7	35_ (7)
PROB_14	35_ (14)
HALITOSE	39A
PROB_7	39_ (7)
PROB_14	39_ (14)
HALITOSE	6A
PDT_D	6D
PDT + PROB_7	6_ (7)
PDT + PROB_14	6_ (14)
HALITOSE	18A
PDT_D	18D
PDT + PROB_7	18_ (7)
PDT + Prob_14	18_ (14)

^1^ Identification of groups that were analyzed in the microbiome.

## Data Availability

Data are available upon request to interested researchers.

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
