# Peer review of "Evaluation of the Oral Microbiome before and after Treatments for Halitosis with Photodynamic Therapy and Probiotics—Pilot Study"

_healthcare, 2024, doi:10.3390/healthcare12111123_

Round 1
Reviewer 1 Report
Comments and Suggestions for Authors
Figures are not readable; the font size, quality, and presentation of the figures must be improved to produce high-quality images.
Moreover, the figure captions should be at the bottom of the figures.
The authors have mentioned in the conclusion the limitations of this work; it is advisable to provide the limitations in the introduction and discussion parts.
Also, while discussing the results, the authors just mean that the findings corroborate with others, but how?
There are many random words like recent, more recent used multiply, and some grammatical errors, like following, that need to be eradicated
Like-383 should change to:
It is an interesting result that the authors could get once they used a group without halitosis, which was not done in our study since we were testing treatment options.
Author Response
Dear editor,
I would like to thank you for the opportunity to submit my article for evaluation at Sensors. I am honored to receive your comments and suggestions to improve my work.
After carefully reviewing the requested changes, we made the necessary changes to meet the magazine's expectations.
I am available to clarify any doubts or discuss any point in question. I hope that the changes made are in line with the magazine's expectations and that my article will be accepted for publication soon.
REVIEWER 1
- Figures are not readable; the font size, quality, and presentation of the figures must be improved to produce high-quality images.
Corrected in text.
- Moreover, the figure captions should be at the bottom of the figures.
Corrected in text.
- The authors have mentioned in the conclusion the limitations of this work; it is advisable to provide the limitations in the introduction and discussion parts.
Corrected in text.
The main limitation of this study, in addition to the number of patients (as it is a pilot study), is the lack of control over the correct oral hygiene of patients at home, despite them having received terms and a folder with information and guidance on hygiene. oral.
- Also, while discussing the results, the authors just mean that the findings corroborate with others, but how?
As for the species, the most prevalent were Neisseria perflava, Streptococcus salivarius, Streptococcus oralis, Veillonella parvula, Granulicatella adiacens, Streptococcus mitis, Granulicatella elegans, Rothia micilaginosa, Gemella sanguis, Streptococcus australis, Schaalia odontolytica, Oribacterium, asaccharolyticum, Streptococcus koreensis tobeensis, Veilon, vestibular Streptococcus, Streptococcus sublava, Fusobacterium periodonticum, campylobacter concisus and Neisseria cirerea. They corroborate with other studies regarding the species Streptococcus mitis, [35,36] Fusobacterium periodonticum [37], Streptococcus oralis, [34] Streptococcus salivarius, [34] Granulicatella elegans [38], as these studies also link halitosis with the presence of these bacteria. The identification of possible bacteria associated with halitosis is essential to develop new strategies in the treatment of halitosis.
- There are many random words like recent, more recent used multiply, and some grammatical errors, like following, that need to be eradicated
Corrected in text.
- Like-383 should change to:
It is an interesting result that the authors could get once they used a group without halitosis, which was not done in our study since we were testing treatment options.
Corrected in text.
Reviewer 2 Report
Comments and Suggestions for Authors
This manuscript investigate oral microbiome before and after treatments for halitosis with photodynamic therapy and probiotics. Overall, the experimental design is reasonable. However, I think the bigger problem of this manuscript is that the sample size of the experimental subjects is very small. The specific comments as follows:
Lack of statistical analysis in the materials and methods section.
All result images are not clear enough and cannot be seen clearly. All images need to be optimized.
The microbiota raw sequencing data need sumitted to the NCBI and provide the accession number.
Line 364-371. All bacterial names at the genus or species level need to be italicized.
Author Response
Dear editor,
I would like to thank you for the opportunity to submit my article for evaluation at Sensors. I am honored to receive your comments and suggestions to improve my work.
After carefully reviewing the requested changes, we made the necessary changes to meet the magazine's expectations.
I am available to clarify any doubts or discuss any point in question. I hope that the changes made are in line with the magazine's expectations and that my article will be accepted for publication soon.
- This manuscript investigate oral microbiome before and after treatments for halitosis with photodynamic therapy and probiotics. Overall, the experimental design is reasonable. However, I think the bigger problem of this manuscript is that the sample size of the experimental subjects is very small.
This manuscript is a pilot project developed to better plan the analyzes that will complement the results.
- Lack of statistical analysis in the materials and methods section.
Corrected in text.
- All result images are not clear enough and cannot be seen clearly. All images need to be optimized.
Corrected in text.
- The microbiota raw sequencing data need sumitted to the NCBI and provide the accession number.
SUB14281977
- Line 364-371. All bacterial names at the genus or species level need to be italicized.
Corrected in text.
Round 2
Reviewer 2 Report
Comments and Suggestions for Authors
This is the second time reviewing this manuscript, and the author has provided corresponding responses and made revisions to the article in response to the review comments on my premise. The current manuscript is generally satisfactory, but there are still some issues that need further revision.
The microbiota raw sequencing data need sumitted to the NCBI and the accession number need add to the manuscript in the Materials and MethodsMaterials and Methods section.
The format of tables in the Figures need to be unified.
Author Response
Dear Reviewer,
Thank you for your considerations.
After carefully reviewing the requested changes, we made the necessary changes to meet the magazine's expectations.
1- The microbiota raw sequencing data need sumitted to the NCBI and the accession number need add to the manuscript in the Materials and Methods section.
The microbiota raw sequencing data was sumitted to the NCBI and the accession number was added to the manuscript in the Materials and Methods section.
https://submit.ncbi.nlm.nih.gov/subs/?search=SUB14281977
SUB14281977
The format of tables in the Figures need to be unified.
The format of tables in the Figures was unified.
I hope that the changes made are in line with the magazine's expectations and that my article will be accepted for publication soon.
Best regards.